# Robust Streaming PCA

**Daniel Bienstock**
IEOR Department
Columbia University

**Minchan Jeong**
Graduate School of AI
KAIST

**Apurv Shukla**
IEOR Department
Columbia University

**Se-Young Yun**
Graduate School of AI
KAIST

{dano,as5197}@columbia.edu, {mcjeong,yunseyoung}@kaist.ac.kr

## Abstract

We consider streaming principal component analysis when the stochastic data-generating model is subject to perturbations. While existing models assume a fixed covariance, we adopt a robust perspective where the covariance matrix belongs to a temporal uncertainty set. Under this setting, we provide fundamental limits on convergence of any algorithm recovering principal components. We analyze the convergence of the noisy power method and Oja's algorithm, both studied for the stationary data generating model, and argue that the noisy power method is rate-optimal in our setting. Finally, we demonstrate the validity of our analysis through numerical experiments on synthetic and real-world dataset.

## 1 Introduction

Principal component analysis (PCA) is one of the most extensively studied methods for obtaining the low-dimensional representation of observed data [22]. However, classical algorithms for PCA store all the observations and use cubic-time complexity, thereby imposing prohibitively large computation-time and space requirements.

Recently, several works on PCA have focused on the design and analysis of streaming algorithms with near-optimal memory and storage complexity [25, 29, 40, 43]. These algorithms assume that all the observations belong to the same low-dimensional space. However, this situation is unlikely when the unknown/unexplored alterations corrupt a system's observations. For instance, it is well known that typical data attacks on power grids can significantly change the estimated covariance matrix of the data observed from sensors [9, 10, 23]. Similarly, PCA can be used to explain stock returns in terms of macroeconomic factors [27], and product pricing taking into account cross-product elasticity and demands [45]. In all these scenarios, the underlying data-generating model changes every instant, and the decisions are based on identifying the changed model.

Current work considers perturbations of the data lying in a *fixed* low-dimensional space [49]. They determine the worst-case position of the adversarial data point to incur the maximum error in the subspace estimated through PCA and measure the distance between the two subspaces using the notion of the principal angle between them. Another line of work considers PCA through the lens of stochastic optimization [5, 42, 51, 52]. Our work differs from these approaches since we assume that the data-generating model changes at every time instant. Further, we propose near-optimal algorithms for recovering principal components under this framework.

We assume a system relies on the time-series of $p$-dimensional vectors sampled from a time-varying model. The available observations are the vectors $(\mathbf{x}_t)_{t=1}^T$. The noisy observation $\mathbf{x}_t \in \mathbb{R}^p$ is a vector lying in the column space of an unobserved full-rank matrix $\mathbf{A}_t \in \mathbb{R}^{p \times k}$. Precisely, from the standard spiked covariance model [35], we consider the time-dependent environment:

$$\mathbf{x}_t \sim \mathcal{N}(\mathbf{0}_{p \times 1}, \, \mathbf{A}_t \mathbf{A}_t^\top + \sigma^2 \mathbf{I}_{p \times p}), \tag{1}$$

---

The authors are ordered alphabetically.

**where $\mathbf{A}_t$ can vary with time.** The parameter $k \ll p$ is the desired number of principal components. Assuming $\mathbf{A}_t$ belongs to a temporal uncertainty set (defined in the equation (2)), our goal is to recover the top-$k$ principal components of the terminal subspace $\mathbf{A}_T$. In financial applications, our model captures the market evolution in terms of the changing $\mathbf{A}_T$ with the ultimate goal of explaining the market conditions for an appropriately chosen $T$.

Previous works on the stationary environment assume $\mathbf{A}_t = \mathbf{A}$ for all $t$ and have focused on computing a basis for the column space of the matrix $\mathbf{A}$ using streaming algorithms [21, 29, 33, 36, 38, 43]. The accurate reconstruction of the principal components for the standard streaming PCA problem depends on the magnitude of observation noise $\sigma$, the dimension of observations $p$, the number of principal components $k$ of the matrix $\mathbf{A}$, and the spectral gap $\delta$ between $k$-th and $k{+}1$-th spectrums. singular value of $\mathbf{A}\mathbf{A}^\top$. In a marked departure from previous work, we study the case when the column space of $\mathbf{A}$ varies across time. This paper explores these avenues and proposes a tractable analysis framework for the streaming PCA problem, robust to perturbations in the data-generating model. Our contributions can be summarized as follows:

1. (Lower Bound; Section 4) Our first contribution is establishing a fundamental lower bound for estimating the principal components when the covariance matrix belongs to a temporal uncertainty set $\mathrm{Tu}(\delta, \Gamma)$. In Theorem 1, we derive the minimax bounds of the expected error for recovering the top singular vectors for any streaming algorithm. Because the underlying distribution can vary, observations from the far past become less important and the estimation of principal components associated with $\mathbf{A}_T$ should be determined by a subset of samples. Simultaneously, it becomes imperative to find the block size $B$ that can be used to recover the principal components. This is in sharp contrast to the standard spiked covariance model. We show that:

   - For $T = \mathcal{O}(\Gamma^{-2/3})$, the minimax estimation error decreases as $\mathcal{O}(p^{1/2}T^{-1/2})$.
   - On the other hand, for $T = \Omega(\Gamma^{-2/3})$, the error stagnates to $\mathcal{O}(p^{1/3}\Gamma^{1/3})$ and does not decrease upon collecting more observations[1].

2. (Algorithm Analysis; Section 5) We then analyze two algorithms to recover the principal components extensively used in the standard streaming PCA setting; the noisy power method [29] and Oja's algorithm [48]. These algorithms represent two very different design principles for computing principal components from data in streaming fashion, processing data in blocks vis-à-vis single observations. We determine the optimal choice for critical parameters: the block size in the noisy power method on Lemma 1 and the learning rate in Oja's algorithm on Lemma 2. Next, we leverage these results to obtain an upper bound on the convergence error for these algorithms in Theorem 2 and 3, respectively. From these results, we have:

   - The block size $B$ for the noisy power method and inverse of the learning rate, $\zeta^{-1}$ for Oja's algorithm, plays a similar role even in the non-stationary environment. In either case, the optimal parameters scale with $\Theta(\Gamma^{-2/3})$, where $\Gamma$ is the perturbation budget.
   - The derived upper bound on the estimation error for the noisy power method matches the minimax error in terms of $p, \delta$, and $\Gamma$ and becomes rate-optimal if it satisfies the mild conditions.

**Notation.** We fix notation throughout the main body of the paper. Matrices are denoted by bold uppercase letters (e.g. $\mathbf{M}$) and vectors are denoted by bold lowercase letters (e.g. $\mathbf{x}$). For $\mathbf{M} \in \mathbb{R}^{p \times k}$, $\mathbf{M}_i$ denote the $i$th column of $\mathbf{M}$, $\mathbf{M}_{i:j}$ denote the submatrix consists with $i$th $\sim j$th column of $\mathbf{M}$ and $\mathbf{M}_{i,j}$ be the $(i,j)$-element of $\mathbf{M}$. $\|\cdot\|$ denote matrix 2-norm or equivalently operator norm for matrices and standard 2-norm for vectors. We use $\mathrm{St}_k(\mathbb{R}^p)$ for the class of orthogonal matrices in $\mathbb{R}^{p \times k}$. $b(\mathbf{M})$ is the orthogonal matrix where the columns form basis for $\mathrm{ran}(\mathbf{M})$. $s_i(\mathbf{M})$ represents the $i$th largest singular value of the matrix.

The singular value decomposition of $\mathbf{M}$ is defined as $\mathrm{SVD}(\mathbf{M}) = \mathbf{U}\mathbf{D}\mathbf{V}^\top$, where $\mathbf{U} \in \mathrm{St}_p(\mathbb{R}^p)$, $\mathbf{V} \in \mathrm{St}_k(\mathbb{R}^k)$, and $\mathbf{D} \in \mathbb{R}^{p \times k}$ is a diagonal matrix whose $i$th diagonal element equals $s_i(\mathbf{M})$. Therefore, we assume without loss of generality that the singular values and respective singular vectors are ordered from largest to smallest. We denote by $\mathbf{M}_\perp$ the orthogonal projection onto the orthogonal complement of the $\mathrm{ran}(\mathbf{M})$. Therefore, if $\mathrm{rk}(\mathbf{M}) = r$ and $\mathrm{SVD}(\mathbf{M}) = \mathbf{U}\mathbf{D}\mathbf{V}^\top$, $\mathbf{M}_\perp$ given by $\mathbf{M}_\perp = \mathbf{I} - \mathbf{U}_{1:r}\mathbf{U}_{1:r}^\top$. Moreover, when $\mathrm{rk}(\mathbf{M}) = \mathrm{rk}(\tilde{\mathbf{M}})$ and $\mathrm{SVD}(\tilde{\mathbf{M}}) = \tilde{\mathbf{U}}\tilde{\mathbf{D}}\tilde{\mathbf{V}}^\top$, the

---

[1]For precise explanation for $\Gamma$ and $\mathrm{Tu}(\delta, \Gamma)$ please refer to the Section 3.

distance between $\mathrm{ran}(\mathbf{M})$ and $\mathrm{ran}(\tilde{\mathbf{M}})$ is defined by:

$$d(\mathrm{ran}(\mathbf{M}), \mathrm{ran}(\tilde{\mathbf{M}})) = \| b(\mathbf{M})b(\mathbf{M})^\top - b(\tilde{\mathbf{M}})b(\tilde{\mathbf{M}})^\top \| = \| \mathbf{U}_{1:k}\mathbf{U}_{1:k}^\top - \tilde{\mathbf{U}}_{1:k}\tilde{\mathbf{U}}_{1:k}^\top \|.$$

We denote $d(\mathrm{ran}(\mathbf{M}), \mathrm{ran}(\tilde{\mathbf{M}}))$ by $d(\mathbf{M}, \tilde{\mathbf{M}})$ whenever clear from the context.

The letter $\mathcal{A}$ stands for abbreviation of sequence of matrices $(\mathbf{A}_t)_{t=1}^T$. We write $\mathcal{X} \sim \mathcal{A}$ when each element $\mathbf{x}_t$ in sequence $\mathcal{X} = (\mathbf{x}_t)_{t=1}^T$ are drawn from $\mathcal{N}(\mathbf{0}, \mathbf{A}_t\mathbf{A}_t^\top + \sigma^2\mathbf{I}_{p\times p})$. We denote the expectation of $f(\mathcal{X})$ over $\mathbf{x}_t \sim \mathcal{N}(\mathbf{0}, \mathbf{A}_t\mathbf{A}_t^\top + \sigma^2\mathbf{I}_{p\times p})$ as $\mathbb{E}_{\mathcal{X}\sim\mathcal{A}}[f]$.

We denote $\tilde{\mathcal{O}}$, $\tilde{\Theta}$ as the $\mathcal{O}$, $\Theta$ notation with ignore the multiplicative dependency of $\log(pT^2)$ or smaller.

## 2 Related Work

Principal component analysis (PCA) has been extensively studied across operations research, computer science, and other communities. We highlight how our work differs from existing literature.

**Robust PCA.** Robust PCA deals with the problem of retrieving the principal components robust to the presence of outliers in the data. The cornerstone work in this direction is the principal component pursuit framework wherein they assume that the matrix of observations $\mathbf{M}$ can be decomposed in terms of a low-rank matrix $\mathbf{L}$ and a sparse matrix (with entries of arbitrarily large magnitude) $\mathbf{S}$ [15].

Several works consider the robust PCA problem and propose algorithms in the offline, batch, and online settings [17, 24, 26, 46]. Our work differs from this line of literature in two aspects: assumptions about the data-generating model and convex optimization techniques. First, our data generation model is unrelated to those considered in the robust PCA literature. Further, rich theories from convex optimization can be used in the PCA framework to design efficient algorithms, but our problem is not amenable to such techniques.

**Streaming PCA.** Streaming algorithms for PCA have been proposed, among other works on PCA [2, 29, 34, 43, 60]. Algorithms analyzed in this work, such as the noisy power method [29] and Oja's algorithm [48], are iterative methods for estimating the principal components. These iterative schemes are instances of stochastic approximation-based solutions for the optimization formulation of the PCA problem [5]. The stochastic approximation is a root-finding framework extensively used for stochastic optimization [11, 37]. Oja's algorithm, originally proposed by [48], was the first such scheme.

This framework is also used to analyze gradient-type and proximal-type incremental methods akin to algorithms for convex optimization. Stochastic gradient descent-based algorithms for the streaming PCA problem, where a single observation is used at every point in time to update the principal components' estimate, have been extensively studied. Along this line, [30] propose GRASTA, an incremental online gradient method for learning over different subspaces. Similarly, [6] proposes GROUSE, based on the idea of gradient updates over the Grassmannian manifold. These and other related works consider Oja's algorithm for the standard streaming PCA problem [1, 16, 31, 31, 53, 59, 62]. However, all these works are based on a completely different modeling assumption than this paper and do not provide theoretical guarantees for our setting.

Streaming and robust PCA algorithms are used in the presence of outliers or data with a lot of missing entries [18, 55]. From this literature, the closest to our work is the work on robust subspace tracking [2, 30, 60]. However, the robustness considered there is against erasures or sparse outliers. While these algorithms provide theoretical guarantees in that setting, those guarantees cannot be extended to our model. Considerations of erasures and outliers under the model proposed in this paper are beyond the scope of this work and remain an interesting future direction.

## 3 Mathematical Framework

Observations from the standard spiked covariance model belong to a fixed $k$-dimensional column space $\mathbf{A} \in \mathbb{R}^p$. Previous work has focused on reconstructing this space from the observed time series. Under our framework, we assume that the sequence of observations $(\mathbf{x}_t)_{t=1}^T$ follows the time-dependent spiked covariance model (1). We consider the problem of computing top-$k$ singular

vectors of $\mathbf{A}_T$. We formulate this model by addressing minimax optimization as a robust optimization problem.

When the adversary is allowed to select a completely arbitrary sequence of matrices $(\mathbf{A}_t)_{t=1}^T$, it is impossible to accurately recover the column space of $\mathbf{A}_T$. Instead, we define temporal uncertainty sets to restrict the power of the adversary.

**Definition 1.** Let $\Gamma, \delta \geq 0$. We only allow the sequence of matrices $(\mathbf{A}_t)_{t=1}^T$ that lie in an temporal uncertainty set $\mathrm{Tu}(\delta, \Gamma)$ defined as:

$$\mathrm{Tu}(\delta, \Gamma) := \left\{ (\mathbf{A}_t)_{t=1}^T \; : \; s_k(\mathbf{A}_t \mathbf{A}_t^\top) \geq \delta \, , \; \|\mathbf{A}_t \mathbf{A}_t^\top - \mathbf{A}_{t-1} \mathbf{A}_{t-1}^\top\| \leq \Gamma, \right\}. \tag{2}$$

We constrain the difference between any two consecutive covariance matrices of the underlying process by $\Gamma$. The assumption $s_k(\mathbf{A}_t \mathbf{A}_t^\top) \geq \delta$ is crucial for establishing bounds of estimation error. It is also justified in any setting where the underlying phenomenology guarantees covariance of rank k; the stated lower bound excludes pathological cases of near-rank $k-1$ or smaller.

Having described the constraints on the perturbation power, we present the algorithms and the performance metric of interest. In particular, we consider **(i)** streaming algorithms, i.e., make a single pass through the time series in chronological order; only the previous samples are stored at any point in the past. These characteristics model the behavior of an algorithm that receives data in real-time and only stores samples in limited memory. Among those streaming algorithms, we consider **(ii)** the algorithms whose output is a set of $k$ orthonormal vectors in $\mathbb{R}^p$. Let us denote by $\Phi$ the family of algorithms just described. The output of algorithm $\phi \in \Phi$ for a given observations $\mathcal{X} = (\mathbf{x}_t)_{t=1}^T$ sampled using model (2) is a set of $k$ orthonormal vectors, which we denote by $\phi_{\mathcal{X}}$. We will also view $\phi_{\mathcal{X}}$ as a matrix in $\mathbb{R}^{p \times k}$ or the subspace generated by $\phi_{\mathcal{X}}$.

Following definition 2 illustrates how we treat the lower bound and when we call the algorithm optimal. We note that this formulation has been widely studied [14, 57, 58].

**Definition 2.** Let $\mathcal{A} = (\mathbf{A}_t)_{t=1}^T \in \mathrm{Tu}(\delta, \Gamma)$, and $\phi \in \Phi$.

1. The estimation error of $\phi$ given $\mathcal{X} \sim \mathcal{A}$ is the distance between the space spanned by $\phi_{\mathcal{X}}$ and the column space of $\mathbf{A}_T$ $(d(\mathrm{ran}(\mathbf{A}_T), \phi_{\mathcal{X}}))$. The metric can be easily extended to the cumulative error. Refer Appendix A for the discussion.

2. $\mathcal{R}^\phi$ is the maximum expected estimation error of $\phi$ under $\mathcal{A}$ over all $\mathcal{A} \in \mathrm{Tu}(\delta, \Gamma)$.

$$\mathcal{R}^\phi := \sup_{\mathcal{A} \in \mathrm{Tu}(\delta, \Gamma)} \mathbb{E}_{\mathcal{X} \sim \mathcal{A}} \big[ d \big( \mathrm{ran}(\mathbf{A}_T), \phi_{\mathcal{X}} \big) \big] .$$

3. $\mathcal{R}^*$ is the minimax estimation error defined as the minimum of the largest expected estimation error incurred by $\phi \in \Phi$:

$$\mathcal{R}^* := \inf_\phi \mathcal{R}^\phi = \inf_\phi \sup_{\mathcal{A} \in \mathrm{Tu}(\delta, \Gamma)} \mathbb{E}_{\mathcal{X} \sim \mathcal{A}} \big[ d \big( \mathrm{ran}(\mathbf{A}_T), \phi_{\mathcal{X}} \big) \big] .$$

4. An algorithm $\phi \in \Phi$ is **rate-optimal** if $\mathcal{R}^\phi \leq C \cdot \mathcal{R}^*$, where constant $C > 0$ is independent of the problem parameters $T$, $\delta$, $p$, $k$, and $\Gamma$.

In this work, we establish the minimax estimation error $\mathcal{R}^*$ and propose rate-optimal sublinear-time, single-pass algorithms for robust streaming PCA.

## 4 Minimax Lower Bound

When $\mathcal{A} = (\mathbf{A}_t)_{t=1}^T$ belongs to the temporal uncertainty set $\mathrm{Tu}(\delta, \Gamma)$ (Definition 1), an algorithm designed to recover the principal components of $\mathbf{A}_T$ from the observations cannot guarantee zero estimation error. Our first goal is to obtain the minimax lower bound on the estimation error in the problem parameters $T$, $\delta$, $p$, $k$, and $\Gamma$.

In order to establish the lower bound, we leverage the fundamental limit of hypothesis tests [57]. The crux of the proof lies in constructing the set of worst-case hypotheses and establishing a lower bound on the probability of error in distinguishing between these hypotheses using observed data. The complete proof is provided in Appendix D.

**Theorem 1** (Lower Bound). *Assume $\delta > \Gamma \geq 0$ and $p > 2k + 1$. For any algorithm $\phi \in \Phi$, there exists a sequence $\mathcal{A} \in \text{Tu}(\delta, \Gamma)$ such that $\mathbb{E}_{\mathcal{X} \sim \mathcal{A}}\big[d\big(\text{ran}(\mathbf{A}_T), \phi_{\mathcal{X}}\big)\big]$ has lower bound with order:*

$$\Theta\left(\min\left\{1, \frac{1}{\sqrt{T}}\left(\frac{p\sigma^2(\sigma^2 + \delta)}{\delta^2}\right)^{1/2} + \left(\frac{\Gamma}{\delta}\right)^{1/3}\left(\frac{p\sigma^2(\sigma^2 + \delta)}{\delta^2}\right)^{1/3}\right\}\right). \tag{3}$$

*By taking $\inf_{\phi \in \Phi}$, we get the same lower bound for $\mathcal{R}^*$.*

For the standard streaming PCA problem (Theorem 1 with the case of $\Gamma = 0$), the fundamental limit is $\Theta((\sigma/\delta)(p(\sigma^2 + \delta)/T)^{1/2})$, which has the expected $\Theta(1/\sqrt{T})$ dependence [14, 58]. On the other hand, in the presence of perturbations ($\Gamma > 0$), the lower bound exhibits a phase transition phenomenon, with the first term representing the effect of model ambiguity. To this end, define the critical time $T_c$ as

$$T_c := \left(\frac{\Gamma}{\delta}\right)^{-2/3}\left(\frac{p\sigma^2(\sigma^2 + \delta)}{\delta^2}\right)^{1/3}. \tag{4}$$

For $T = \mathcal{O}(T_c)$, the lower bound decreases with the rate of $1/\sqrt{T}$. However, when $T = \Omega(T_c)$, the first term dominates the second term, and $\mathcal{R}^*$ becomes independent of the number of observations $T$. In this regime, the error stagnates to $\mathcal{O}((\Gamma/\delta)^{1/3}\left(p\sigma^2(\sigma^2 + \delta)/\delta^2\right)^{1/3})$. Therefore, as our intuition suggests, the information quickly becomes stale in a dynamic environment.

Theorem 2 and 3 will prove that the noisy power method and Oja's algorithm attain a near-optimal bound on the convergence guarantee. Theorem 2 guarantees that if $s_1(\mathbf{A}_t \mathbf{A}_t^\top) = \Theta(\delta)$, the upper bound for estimation error on the noisy power method is of the following order:

$$\mathcal{R}^{\text{NPM}} = \tilde{\mathcal{O}}\left(\left(\frac{\Gamma}{\delta}\right)^{1/3}\left(\frac{(p\sigma^2 + k\delta)(\sigma^2 + \delta)}{\delta^2}\right)^{1/3}\right). \tag{5}$$

That is, if $p\sigma^2$ dominates $k\delta$, the noisy power method becomes rate-optimal under the controlled uncertainty with $\text{Tu}(\delta, \Gamma)$. This regime is the case of noisy practical situations, with $\sigma^2 \not\ll \delta$.

## 5 Convergence Analysis

In this section, we analyze two algorithms for the robust streaming PCA problem. A generic template for algorithms, $\phi \in \Phi$ of interest to us is as follows: **(i)** $\phi$ is initialized with a random matrix with orthonormal columns $\hat{\mathbf{U}} \in \mathbb{R}^{p \times k}$; **(ii)** a running estimate of the principal components is maintained as the columns of $\hat{\mathbf{U}}$; **(iii)** observations are projected onto the column space of $\hat{\mathbf{U}}$ to update this estimate.

We consider two algorithms: a robust version of the noisy power method (Algorithm 1) and Oja's algorithm (Algorithm 2). The critical difference between the noisy power method and Oja's algorithm is the data used to estimate the principal components. In the noisy power method, the estimates are updated after a batch of observations, whereas in Oja's algorithm, the estimates are updated after scaling every observation with the learning rate. Therefore, the parameters determining the performance of these algorithms are the batch size $B$ for the robust power method and the learning rate $\zeta$ for Oja's algorithm.

The analysis of these algorithms cannot be readily established with existing techniques when the covariance matrix belongs to a temporal uncertainty set since they rely on showing that the estimates improve every iteration. Further, applying many concentration results requires random matrices to be bounded, which is not the case when the observations are sampled from (1). Therefore, in order to simplify the analysis of both algorithms, we introduce Assumption 1, adapted from [34].

**Assumption 1.** Let $(\mathbf{A}_t)_{t=1}^T \in \text{Tu}(\delta, \Gamma)$. For $\tilde{\delta} \geq \delta$ and $\mathcal{M}, \mathcal{V} > 0$, we consider the observations $\mathbf{x}_t$ for $t \in [T]$ satisfy the following:

1. $\mathbb{E}[\mathbf{x}_t \mathbf{x}_t^\top] = \mathbf{A}_t \mathbf{A}_t^\top + \sigma^2 \mathbf{I}_{p \times p}$ while $\|\mathbf{A}_t \mathbf{A}_t^\top\| \leq \tilde{\delta}$,

2. $\|\mathbf{x}_t \mathbf{x}_t^\top - (\mathbf{A}_t \mathbf{A}_t^\top + \sigma^2 \mathbf{I}_{p \times p})\| \leq \mathcal{M}$ a.s., and

3. $\left\|\mathbb{E}\left[(\mathbf{x}_t \mathbf{x}_t^\top - (\mathbf{A}_t \mathbf{A}_t^\top + \sigma^2 \mathbf{I}_{p \times p}))^2\right]\right\| \leq \mathcal{V}$.

When the observations follow model (1), we condition our analysis on the high-probability event $\mathfrak{E}$.

**Definition 3.** Let $\mathrm{SVD}(\mathbf{A}_t\mathbf{A}_t^\top + \sigma^2\mathbf{I}_{p\times p}) = \mathbf{U}_t\mathbf{D}_t\mathbf{U}_t^T$ and $\mathbf{x}_t = \mathbf{U}_t\mathbf{D}_t^{1/2}\mathbf{z}_t$, where $\mathbf{z}_t \sim \mathcal{N}(\mathbf{0}, \mathbf{I}_{p\times p})$. We define the event $\mathfrak{E}$ as:

$$\mathfrak{E} := \left\{ \mathbf{z}_t \in \left[ -\sqrt{2\log(2pT^2)}, \sqrt{2\log(2pT^2)} \right]^p ; \forall t \in [T] \right\}.$$

The observations from model (1) satisfy Assumption 1 with $\mathcal{M} = 2(k\tilde{\delta} + p\sigma^2)(1 + \Theta(\log(pT^2)/T))$, $\mathcal{V} = 2\mathcal{M}(\tilde{\delta} + \sigma^2)$ with probability $\mathbb{P}[\mathfrak{E}] \geq 1 - 1/T$. Although after conditioning $\mathbb{E}[\mathbf{x}_t\mathbf{x}_t^\top|\mathfrak{E}] \neq \mathbf{A}_t\mathbf{A}_t^\top + \sigma^2\mathbf{I}_{p\times p}$, we can use all the results in Section 5.1 and 5.2 with a multiplicative logarithmic factor. Please refer to Appendix B for the details.

## 5.1 Noisy Power Method

---

**Algorithm 1** Noisy Power Method with block size $B$ [29]

---

1: **Input:** Stream of vectors: $(\mathbf{x}_t)_{t=1}^T$, block size: $B$, dimensions: $p, k$
2: Sample each element of $\hat{\mathbf{U}}(0)$ in $\mathcal{N}(0, 1)$
3: **for** $\ell = 1 : L = \lfloor T/B \rfloor$ **do**
4:     $\mathbf{Y}(\ell) \leftarrow \mathbf{0} \in \mathbb{R}^{p\times k}$
5:     **for** $t = (\ell-1)B + 1 : \ell B$ **do**
6:         $\mathbf{Y}(\ell) \leftarrow \mathbf{Y}(\ell) + \frac{1}{B}\mathbf{x}_t\mathbf{x}_t^\top\hat{\mathbf{U}}(\ell-1)$
7:     **end for**
8:     $\hat{\mathbf{U}}(\ell) \leftarrow \mathrm{Gram\text{-}Schmidt}(\mathbf{Y}(\ell))$
9: **end for**
10: **Output:** $\hat{\mathbf{U}}(L)$

---

The noisy power method is an iterative algorithm for computing the top-$k$ principal components of a matrix. Starting from the random matrix $\hat{\mathbf{U}}(0)$ in $\mathbb{R}^{p\times k}$, the algorithm runs for $L$ iterations, each processing $B$ samples. By repeating this procedure, we expect the algorithm to reconstruct the covariance matrix if the observations are derived from the fixed distribution.

When observations are drawn from model (1) under Assumption 1, the later observation can be sampled from distributions with shifted covariance. Unlike the standard case, for any algorithm $\phi \in \Phi$, the presence of perturbation prevents the convergence of the columns of $\hat{\mathbf{U}}(\ell)$ to the singular vectors of the final covariance matrix $\mathbf{U}(\ell)$. The main difficulty here is that the columns of $\hat{\mathbf{U}}(\ell)$ do not converge towards a fixed set of vectors but keep tracking the time-varying principal components. Recall that our ultimate objective is to recover the principal components associated with the *terminal* observation. Hence, we decompose the covariance matrix in terms of the last observation and the remaining samples. For $\ell$-th block, we denote the covariance matrix for $\ell B$-th observation as $\mathbf{M}(\ell)$ and have:

$$\frac{1}{B}\sum_{t=(\ell-1)B+1}^{\ell B}\mathbf{x}_t\mathbf{x}_t^\top = \mathbb{E}[\mathbf{x}_{\ell B}\mathbf{x}_{\ell B}^\top] + \mathcal{E}(\ell) = \underbrace{\mathbf{A}_{\ell B}\mathbf{A}_{\ell B}^\top + \sigma^2\mathbf{I}}_{\mathbf{M}(\ell)} + \mathcal{E}(\ell). \tag{6}$$

Due to perturbations in the robust model, $\mathcal{E}(\ell)$ is a non-zero mean random variable. Therefore, we first decompose $\mathcal{E}(\ell)$ in terms of the contribution due to inherent noise and the perturbations the robust model allows. Then, in Lemma 1, we decompose the error $\|\mathcal{E}(\ell)\|$ with respect to the block size $B$ and the allowed perturbations $\Gamma$ in the robust model. We provided complete proof in Appendix E.

**Lemma 1** (Spectral norm of noise). *Assume that the observations $(\mathbf{x}_t)_{t=1}^T$ generated according to the Assumption 1. With probability greater than $1 - 1/T$, the matrices $\mathcal{E}(\ell)$ in the equation (6) are bounded by:*

$$\max_{1\leq\ell\leq L}\|\mathcal{E}(\ell)\| \leq \frac{1+3\sqrt{2}}{3}\sqrt{\frac{\mathcal{V}\log(2pT^2)}{B}} + \frac{B\Gamma}{2}. \tag{7}$$

*when the block size $B$ is larger than $\mathcal{M}^2\log(2pT^2)/\mathcal{V}$.*

Lemma 1 highlights the effect of allowing perturbations in the data generation model. From classical results in statistics, our intuition tells us that the effect of noise washes out as the block size increases, i.e., the error decays with the 'inverse square root' of the block size. Hence, barring memory and

data issues, a larger block size is better when $\Gamma = 0$. In contrast, covariance perturbations add errors proportional to the block size. Consequently, we have a trade-off between two terms in this case, and an optimal block size exists depending on the $\Gamma$.

We establish the convergence guarantee of the robust power method in Theorem 2. In the proof of Theorem 2 (Appendix F), we bound the distance, $d\big(\mathbf{U}(L), \hat{\mathbf{U}}(L)\big)$ between the output of Algorithm 1, $\hat{\mathbf{U}}_{1:k}(L)$ and $k$-orthonormal vectors spanning the column space of $\mathbf{A}_T$, $\mathbf{U}_{1:k}(L)$. We identify the optimal block size $B$, the unique parameter for the noisy power method, and establish an upper bound on the estimation error of the noisy power method.

**Theorem 2** (Robust power method). *Assume that $\delta \geq 0.71\sigma^2$ and $\Gamma = \mathcal{O}(\delta^3/(\mathcal{V}\log(2pT^2)))$. When the observations $(\mathbf{x}_t)_{t=1}^T$ satisfies the assumption 1, for $B = \Theta(\mathcal{V}^{1/3}\log(2pT^2)^{1/3}/\Gamma^{2/3})$ we have:*

$$d(\mathrm{ran}(\mathbf{A}_T), \mathrm{NPM}_{\mathcal{X}}) = \mathcal{O}\left(\frac{(\mathcal{V}\Gamma\log(2pT^2))^{1/3}}{\delta} + 0.7^{T/B}\right), \tag{8}$$

*with probability $1 - 2/T - c^{\Omega(p-k+1)} - e^{-\Omega(p)}$.*

When $\Gamma = \Omega(\delta^3/(\mathcal{V}\log(2pT^2)))$ we have that $(\mathcal{V}\Gamma\log(2pT^2))^{1/3}/\delta = \Omega(1)$. Therefore, the condition on $\Gamma$ in Theorem 2 is necessary to avoid a trivial upper bound $\Theta(1)$. This condition encompasses several applications of interest alluded to earlier. For example, in the financial applications alluded to earlier, individual market changes of interest happen on a millisecond time scale. It is of significant interest to terminally detect incremental market changes. Our results hold on to the large value of cumulative changes and allow us to study them. They further imply that the noisy power method is rate-optimal for non-trivial values of $\Gamma$. When the observations follow model (1), under the event $\mathfrak{E}_\Delta$, Theorem 2 shows that the robust power method can achieve an estimation error of:

$$\mathcal{R}^{\mathrm{NPM}} = \tilde{\mathcal{O}}\left(\left(\frac{\Gamma}{\delta}\right)^{1/3}\left(\frac{(p\sigma^2 + k\delta)(\sigma^2 + \delta)}{\delta^2}\right)^{1/3}\right), \tag{9}$$

if $T = \Omega(\max(T_c, \delta(p\sigma^2)^{-1}))$, $\Gamma = \Omega((c^{\Omega(p-k+1)} + e^{-\Omega(p)})\delta^2(p\sigma^2)^{-1})$, and $\tilde{\delta} = \Theta(\delta)$.

Then it becomes order-wise identical to the fundamental limit established in Theorem 1 when $p\sigma^2$ dominates $k\delta$. The first condition on $T$ illustrates when past observations become less critical. The probability for the upper bound on the noisy power method [29] with random initialization should not be small to construct expectation bounds from the high probability bound. We address this regime by condition on $\Gamma$, which is coarse due to exponential terms and $(p\sigma^2)^{-1}$.

Establishing bounds on the estimation error when the underlying singular vectors change is intricate since the subspace to which consecutive observations belong is potentially different. Conventional proofs that analyze noisy power methods or Oja's algorithm show that under a variety of assumptions at every iteration $\ell$, the distance between the estimated and true subspace, $d(\mathbf{U}(\ell), \hat{\mathbf{U}}(\ell))$ decreases. For instance, the proof in [29] requires $\|\mathcal{E}(\ell)\mathbf{U}(\ell)\| = \mathcal{O}(\sqrt{k/p})$, which does not hold under our model since $\|\mathbf{M}(\ell) - \mathbf{M}(\ell-1)\|$ is, in general, greater than $\sqrt{k/p}$. Similarly, the concentration approach in [33] can be used only when the covariance matrix is time-invariant. We briefly describe our proof technique to establish Theorem 2, deferring details to Appendix F. Let $\mathcal{M}^{(L)}$ denotes the product $\prod_{\ell=1}^L \big(\mathbf{M}(\ell) + \mathcal{E}(\ell)\big)$. Then, the output of the algorithm $\hat{\mathbf{U}}_{1:k}(L)$ is an orthonormal basis of $\mathcal{M}^{(L)}\hat{\mathbf{U}}(0)$, which estimates the first $k$ principal components $\mathbf{U}_{1:k}$ of the $\mathbf{M}(L)$. To address this, we construct sequences of $k$ and $(p-k)$-dimensional subspaces of $\mathbb{R}^p$ from observations $\{(\mathbf{x}_t)_{t=(\ell-1)B}^{\ell B}\}_{\ell=1}^L$, denoted by $\{\mathbf{N}^{(\ell)}\}_{\ell=1}^L \in \mathrm{St}_{p-k}(\mathbb{R}^p)$ and $\{\mathbf{W}^{(\ell)}\}_{\ell=1}^L \in \mathrm{St}_k(\mathbb{R}^p)$ respectively, such that for all iterations $\ell$:

1. $\mathrm{ran}\big((\mathbf{M}(\ell)+\mathcal{E}(\ell))\mathbf{N}^{(\ell)}\big) \subseteq \mathrm{ran}\big(\mathbf{N}^{(\ell+1)}\big)$ and $\mathrm{ran}\big((\mathbf{M}(\ell)+\mathcal{E}(\ell))\mathbf{W}^{(\ell)}\big) \subseteq \mathrm{ran}\big(\mathbf{W}^{(\ell+1)}\big)$,

2. $\|(\mathbf{M}(\ell) + \mathcal{E}(\ell))\mathbf{N}^{(\ell)}\| \cdot \|((\mathbf{M}(\ell) + \mathcal{E}(\ell))\mathbf{W}^{(\ell)})^{-1}\| < 1$,

3. $d(\mathbf{U}_{1:k}(\ell), \mathbf{W}^{(\ell+1)}), d(\mathbf{U}_{k+1:p}(\ell), \mathbf{N}^{(\ell+1)}) = \mathcal{O}(\|\mathcal{E}(\ell)\|)$.

The initial random matrix $\hat{\mathbf{U}}_{1:k}(0)$ consists of both $\mathbf{N}^{(1)}$ and $\mathbf{W}^{(1)}$ with high probability. From the first two properties, at every iteration $\ell$, the projection of $\hat{\mathbf{U}}_{1:k}(\ell-1)$ in $\mathbf{W}^{(\ell)}$ is amplified more than

that on $\mathbf{N}^{(\ell)}$ and thus $\hat{\mathbf{U}}_{1:k}(L)$ becomes very close to $\mathbf{W}^{(L)}$ after sufficiently large $L$. From the last property, we can conclude that $\mathbf{W}^{(L)}$ is close to $\mathbf{U}_{1:k}(L)$, where the distance between $\mathbf{W}^{(L)}$ and $\mathbf{U}_{1:k}(L)$ is proportional to $\|\mathcal{E}(L)\|$. Combining these ideas with Lemma 1 establishes the results.

## 5.2  Oja's Algorithm

---

**Algorithm 2** Oja's Algorithm with learning rate $\zeta$ [48]

---

1: **Input:** Stream of vectors: $(\mathbf{x}_t)_{t=1}^T$, learning rate: $\zeta$, and dimensions: $p, k$
2: Sample each element of $\hat{\mathbf{U}}(0)$ in $\mathcal{N}(0, 1)$
3: **for** $t = 1 : T$ **do**
4:    $\hat{\mathbf{U}}(t) \leftarrow$ Gram-Schmidt$((\mathbf{I}_{p \times p} + \zeta \mathbf{x}_t \mathbf{x}_t^\top) \hat{\mathbf{U}}(t-1))$
5: **end for**
6: **Output:** $\hat{\mathbf{U}}(T)$

---

We now establish the convergence guarantee for Oja's Algorithm (Algorithm 2) when observations follow the equation (1). Unlike the noisy power method, Oja's Algorithm is multiplicative in its construction of the estimated subspace. We extend the existing analysis for Oja's algorithm [3, 33, 34] by considering a virtual block with $B = B_\zeta = \zeta^{-1}$ observations. Building upon the analysis framework for the noisy power method and intuition from binomial approximation $(1 + \zeta x)^{1/\zeta} \simeq (1 + x)$ we establish the convergence guarantees for Oja's algorithm. Like the previous section, we decompose the block with target $\mathbf{M}(\ell)$ and error matrix $\mathcal{E}'(\ell)$ as:

$$\prod_{t=(\ell-1)B+1}^{\ell B} (\mathbf{I}_{p \times p} + \zeta \, \mathbb{E}[\mathbf{x}_t \mathbf{x}_t^\top]) = \prod_{t=(\ell-1)B+1}^{\ell B} (\mathbf{I}_{p \times p} + \zeta \, \mathbb{E}[\mathbf{x}_{\ell B} \mathbf{x}_{\ell B}^\top]) + \mathcal{E}(\ell) = \mathbf{M}_{\mathrm{Oja}}(\ell) + e^{\tilde{\delta}+\sigma^2} \mathcal{E}'(\ell) . \quad (10)$$

where $\mathcal{E}'$ is scaled error matrix. In Lemma 2, we provide bound for scaled error matrix with respect to the learning parameter $\zeta$, or the virtual block size $B_\zeta = \zeta^{-1}$ similar to Lemma 1. The proof is provided in Appendix G.

**Lemma 2** (Spectral norm of noise, Oja's algorithm case)**.** *Assume that the observations $(\mathbf{x}_t)_{t=1}^T$ generated according to the Assumption 1. With probability greater than $1 - 1/T$, the matrices $\mathcal{E}'(\ell)$ on the equation (10) are bounded by:*

$$\max_{1 \le \ell \le L} \|\mathcal{E}'(\ell)\| \le \sqrt{\frac{2e^2 \mathcal{M}^2 \log(pT^2)}{B_\zeta}} + \frac{B_\zeta \Gamma}{2} + \mathcal{O}(B_\zeta \Gamma) , \quad (11)$$

*when the virtual block size $B_\zeta = \zeta^{-1}$ is larger than $2\mathcal{M}^2 \log(pT^2)$.*

The difference in parameter $\mathcal{M}$, rather than $\mathcal{V}$, as in the case of Lemma 1 arises due to the use of a different concentration inequality. The estimator of the noisy power method averages the outer product of vectors (equation (10)), making it straightforward to use Bernstein's inequality. On the other hand, Oja's algorithm averages the product of random matrices rather than the sum of random matrices. Therefore, we introduce the multiplicative concentration inequality [32], which requires the (probabilistic) norm bound for matrices. Combining multiplicative concentration inequalities with Lemma 2 and the techniques developed for the noisy power method, we obtain a convergence guarantee for Oja's algorithm in Theorem 3. The complete proof is provided in Appendix H.

**Theorem 3** (Oja's algorithm)**.** *Assume that $\delta \ge 0.71$ and $\Gamma = \mathcal{O}(\delta^3/(e^{3(\tilde{\delta}-\delta)} \mathcal{M}^2 \log(pT^2)))$. When the observations $\{\mathbf{x}_t\}_{t=1}^T$ satisfies the assumption 1, for $\zeta^{-1} = \Theta(\mathcal{M}^{2/3} \log(pT^2)^{1/3}/\Gamma^{2/3})$ we have:*

$$d(\mathrm{ran}(\mathbf{A}_T), \mathrm{Oja}_\mathcal{X}) = \tilde{\mathcal{O}}\left( e^{\tilde{\delta}} \frac{(\mathcal{M}^2 \Gamma \log(pT^2))^{1/3}}{\delta} + 0.7^{\zeta T} \right) , \quad (12)$$

*with probability $1 - 2/T - c^{\Omega(p-k+1)} - e^{-\Omega(p)}$.*

Unlike the noisy power method, the upper bound in Theorem 3 is $\mathcal{O}(p^{2/3})$ rather than the optimal-dependence of $\mathcal{O}(p^{1/3})$ from Theorem 1. It is unclear whether the upper bound can be improved. Sharpening our analysis with a two-phase strategy [33, 39] (wherein the first phase identifies a good initial point and the second phase establishes convergence given an initial point) might be an excellent direction for future investigation.

# 6 Numerical Results

Key observations from Theorem 2 and 3 on each algorithm illustrate; **(i)** the existence of the optimal block size $B$ and the learning rate $\zeta$ to obtain the minimum recovery error, and **(ii)** $\Gamma^{-3/2}$ dependencies of that optimal $B$ and $1/\zeta$. In order to verify the established results for both algorithms, this section provides the performance of algorithms for various environments. We synthesized the $\mathcal{A} = (\mathbf{A}_t)_{t=1}^T$ and sample $\mathcal{X} = (\mathbf{x}_t)_{t=1}^T$ from $\mathcal{A}$. We generate $(\mathbf{A}_t)_{t=1}^T \in \mathbb{R}^{p \times k}$ as the product of three matrices, $\mathbf{U}_t \in \mathrm{St}_p(\mathbb{R}^p)$, $\mathbf{D}_t \in \mathbb{R}^{p \times k}$ (; diagonal), and $\mathbf{V}_t \in \mathrm{St}_k(\mathbb{R}^k)$. To obtain the matrix of the next step, we rotate the first matrix as $\mathbf{U}_t = \mathbf{U}_{t-1}\mathbf{R}_t$ ($\mathbf{R}_t \in \mathrm{SO}(p)$). The vectors $\mathbf{x}_t$ are sampled from the model (1). More details for experimental setup are described in Appendix I.

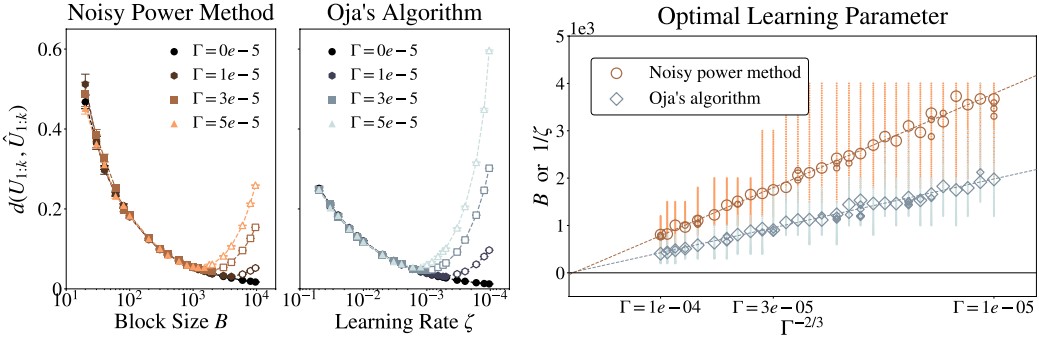

(a) Distance between true column space of $\mathbf{A}_T$ (; $\mathbf{U}_{1:k}$) and estimated space at $t = T$ (; $\hat{\mathbf{U}}_{1:k}$) as $\Gamma$ varies.  (b) Empirical optimal block size $B$ and learning rate $\zeta$ as $\Gamma$ varies.

Figure 1: Numerical results with synthesized streams of vectors. We used the setting $(\sigma, \delta, p, k) = (0.15, 1.0, 100, 5)$.

Our first observation in Figure 1a is that convergence error decreases as the block size increases without any covariance perturbation ($\Gamma = 0$). This behavior is expected since increasing the number of past information results in better accuracy guarantees for the recovered space in the absence of a covariance shift. However, if the covariance perturbation exists ($\Gamma > 0$), the optimal learning parameter exists, and we have the smaller optimal block size with stronger perturbation. This phenomenon is also expected since an increase in the adversarial budget implies that the past information becomes less relevant. Our observations also corroborate our theoretical results in Theorem 1.

In Figure 1b, we focus on the optimal value of the block size $B$ and the inverse of the learning rate $1/\zeta$ and its variation with the perturbation budget $\Gamma$. We plot the empirically optimal learning rate for the case of the noisy power method and Oja's algorithm with $\Gamma^{-2/3}$. We observe that the optimal block size and the inverse learning rate are proportional to $\Gamma^{-2/3}$. This experimental dependency of $\Gamma^{-2/3}$ verifies the theoretically prescribed results in Theorem 2 and 3.

## 6.1 Experiments on Stock Price Dataset

We provide the real-world benchmark using the S&P500 stock market dataset [44] in Kaggle to test our findings in the non-stationary environment. Refer the Appendix J for the non-stationarity of environments and further experimental details that do not appear in the main paper.

The stream of vectors consisted of $133 (= p)$ companies' normalized daily returns. Since each company in S&P500 has a different time horizon of available information, we considered 133 companies with the cost information from Mar.18, 1980, to Jul.22, 2022 ($T = 10677$). Then we calculated the 'daily return,' which is the difference of adjusted close cost between two days; normalized by the adjusted close cost of the day. The stream of vectors from the environment can be seen as sampled from time-varying distributions with $\Gamma \simeq 0.17$.

The objective is to predict the principal components of the covariance matrix of daily returns as in [4, 54, 61]. We tested the noisy power method and Oja's algorithm on the preprocessed stream. For the target space $\mathbf{A}_T$, we used the $k$-dimensional subspace consisting of top-k singular vectors of covariance estimator calculated with the final 500 samples ($k = 1, \ldots, 5$).

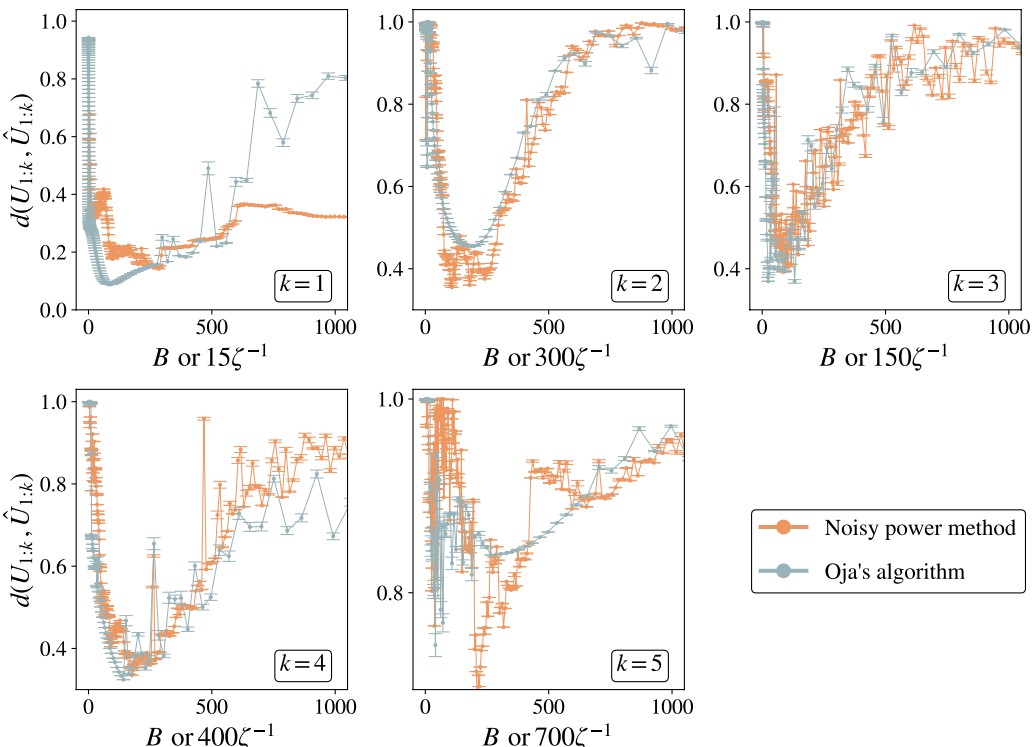

Figure 2: S&P500 stock market daily return prediction, for $k = 1, 2, \ldots, 5$.

The results in Figure 2 indicate that our findings are also valid in the real-world environment with covariance shifts. First, each result shows optimal parameters regimes observed at the U-shaped curves on recovery errors. Furthermore, the result on the noisy power method with varying block size $B$ is akin to the recovery error on Oja's algorithm, plotted with scaled inverse learning rate $\zeta^{-1}$. These two observations support the main findings in Figure 1a. We also note that different scaling for each k is natural since we have different spectral gaps $\delta$ between $k$-th and $k+1$-th spectrums.

## 7  Conclusion

On the streaming PCA settings with time-varying covariance, we analyzed the fundamental lower bound of the minimax error and estimation errors for the noisy power method and Oja's algorithm. Under this framework, when no perturbation exists ($\Gamma = 0$), our theoretical result on a lower limit coincides with the order of the traditional outcome. Furthermore, when perturbation exists ($\Gamma > 0$), we have a non-avoidable positive minimax recovery error, although the time horizon becomes arbitrarily long. Next, we found that the noisy power method order-wisely achieves this recovery error and becomes rate-optimal if the $\sigma^2 = \Omega(k\delta/p)$. For the optimal learning parameters $B$ or $\zeta^{-1}$, we showed that optimal block size and inverse learning rate minimizing recovery error are similar up to a multiplicative factor and proportional to $\Gamma^{-3/2}$. Experimental results both on the synthetic data and real-world environments support the theoretical findings on the learning parameters.

Although our analysis requires the spectral gap assumption, a recent line of literature considers effective rank [12, 50] as a crucial metric. Therefore, establishing guarantees for robust streaming PCA in terms of effective rank is a perfect direction for future work.

## Acknowledgement

DB and AS were supported by a DARPA Lagrange award. MJ and SY were supported by Institute of Information & Communications Technology Planning & Evaluation (IITP) grant funded by the Korea government(MSIT) (No.2022-0-00311, Development of Goal-Oriented Reinforcement Learning Techniques for Contact-Rich Robotic Manipulation of Everyday Objects; No.2019-0-00075, Artificial Intelligence Graduate School Program(KAIST)).

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
