# OpenReview forum: "Robust Streaming PCA"
_NeurIPS.cc/2022/Conference — NeurIPS 2022 Accept_

### Official Review · Reviewer_3FNv · 2022-07-01

**Rating:** 6
**Confidence:** 3
**Soundness:** 3 good
**Presentation:** 3 good
**Contribution:** 2 fair

**Summary:**

The authors analyze the convergence of the noisy power method and Oja’s algorithm in the field of robust streaming PCA.

**Questions:**

Can the authors  precise if they tested the algorithms in the presence of outliers or partly observed data?

**Limitations:**

The authors have well addressed the limitations of their work.

**Strengths And Weaknesses:**

Strengths
1) The paper is well  organized.
2) This study about the fundamental limit of the minimax error and convergence of the noisy power method and Oja’s algorithm is welcome and well conducted.

Weaknesses:
1) There are missing references about key works in the field of streaming PCA. Please see:
L Balzano, “On the equivalence of Oja's algorithm and GROUSE”, International Conference on Artificial Intelligence and Statistics, AISTATS 2022, Valencia, Spain, March 2022.
Oja's improvements:
K. Abed-Meraim, S. Attallah, A. Chkeif, Y. Hua, “Orthogonal Oja algorithm", IEEE Signal Processing Letters, Volume 7, no. 5, pages 116-119, 2000.
S. Wu, H. Wai, A. Scaglione, N. Jacklin, “The Power-Oja method for decentralized subspace estimation/tracking”, IEEE International Conference on Acoustics, Speech and Signal Processing, ICASSP 2017, pages 3524-3528, 2017.
PAST/OPAST
B. Yang, “Projection approximation subspace tracking", IEEE Transactions on Signal Processing, Volume 43, pages 95–107, January 1995.
K. Abed-Meraim, A. Chkeif, Y. Hua, “Fast orthonormal PAST algorithm,” IEEE Signal Processing Letters, Volume 7, No. 3, pages60–62, 2000.
PETRELS and its improvements
Y. Chi, R. Calderbank, Y. Eldar, “PETRELS: Parallel subspace estimation and tracking by recursive least squares from partial observations", IEEE Transactions on Signal Processing, Volume 61, No. 23, pages 5947–5959, 2013.
N. Linh-Trung, V. Nguyen, M. Thameri, T. Minh-Chinh, K. Abed-Meraim, "Low-complexity adaptive algorithms for robust subspace tracking", IEEE Journal of Selected Topics in Signal Processing, Volume 12, No. 6, pages 1197-1212, 2018.
L. Thanh, N. Dung, N. Linhtrung, K. Abed Meraim, "Robust Subspace Tracking with Missing Data and Outliers: Novel Algorithm with Convergence Guarantee", IEEE Transactions on Signal Processing, 2021.

2) The authors missed to indicate in one or two sentence that streaming algorithms attempt to deal either or both the presence of outliers or partly observed data.

---

> ### Author Response · Authors · 2022-08-02
> **Response to Reviewer 3FNv**
>
> We thank the reviewer for the careful reviews. We would like to address each of the two issues raised by the reviewers below.
>
> #### **Citation Issue**
> We thank the reviewer for highlighting these essential references in the relative fields. As suggested, we have included the references in the new version of the manuscript. For this issue, please refer to the common rebuttal ('Providing more related works on the fields and citations'). We have mentioned the usage of streaming algorithms for comparison to this line of algorithms (Line 125-130).
>
> #### **Our work on the presence of outliers or partly observed data - Experimental results**
> We appended the experiment with the real-world stock price dataset. The results support one of the most valuable findings: the optimal learning parameter in the non-stationary setting, and block size (for noisy power method) and inverse learning rate (for Oja's algorithm) play a similar role in the non-stationary environment. The stock price dataset inherits outliers by sudden events, and a non-negligible proportion of data is missing. We expect this new result to resolve the reviewer's question. Please refer to the common rebuttal at the top and Appendix J.
>
> #### **Our work on the presence of outliers or partly observed data - Theoretical View**
> As the reviewer pointed out, our theoretical analysis does not explicitly consider the sparse outliers and masking. Ours is the first work to consider time-varying covariance and consequently, issues of erasures and outliers is beyond our scope and would be an interesting future work. We have remarked upon this in the revised version of the manuscript.

---

### Official Review · Reviewer_pYST · 2022-07-09

**Rating:** 6
**Confidence:** 3
**Soundness:** 3 good
**Presentation:** 4 excellent
**Contribution:** 3 good

**Summary:**

This paper studies the properties of streaming PCA in a dynamic environment.  In particular, the authors consider the problem of recovering the top k eigenvectors of the last observation in the stream even when the associated covariance matrices are drifting.  The authors establish a minimax lower bound for the problem and show that the iterated power method achieves the minimax rate.  They also derive convergence rates for Oja’s algorithm in this setting, another commonly used method for streaming PCA.


**Questions:**

- Some recent work has emerged on PCA where the problem complexity is captured by the effective rank, defined as $r(\Sigma) = tr(\Sigma)/ \lambda_1(\Sigma)$ instead of the dimension.  Do you believe that some of your analysis will be useful for deriving minimax rates in this setup?  Either way, I believe that this line of work should be cited and mentioned briefly.
- The Gaussian spiked covariance model is a bit more restrictive than other setups considered in the literature (although these papers consider an IID stream).  Could you discuss how much the minimax results depend on the Gaussianity assumption?  It seems that the assumption is not needed in for deriving upper bounds for the streaming algorithms.


**Limitations:**

Yes, the authors are honest about the suboptimality of the rate for Oja's algorithm.  It may also be worthwhile to state that minimax results may be pessimistic regarding the difficulty of the problem when more structure is present.

**Strengths And Weaknesses:**

Strengths:
- The authors address an interesting problem that, to the best of my knowledge, has not been studied very much: streaming PCA with time-varying covariance matrices.
- The authors consider the problem of recovering the top k eigenvectors opposed to the the top eigenvector, which has only recently started to be addressed in some generality in the streaming PCA literature.
- The minimax rate is a nice result that characterizes the difficulty of the problem and also quantifies the effect of drift well.
- The authors show that a commonly used streaming method (noisy power method) attains the minimax rate and also provide analysis for another widely used algorithm (Oja’s algorithm).
- On the technical side, the time-varying nature of the problem leads to additional complications and more involved/novel analysis.

Weaknesses:
- To me, it appears that this time-varying covariance matrix setting is novel but somewhat unnatural. I believe that the authors should provide better motivation for their problem.  Arguably, the top k eigenvectors of the expectation of the sample covariance is a more natural target than the eigenspace for the last element in the data stream.  Under mild perturbations, one would expect that the mean of this expectation would still converge to some limit and is a meaningful target.  One would also expect that that recovering this eigenspace is much easier.
- While the spiked covariance matrix is a common setup, minimax rates for this class have rather poor dependence on dimension.  If one solely considers regimes suggested by these bounds, then one of the main benefits of streaming methods, namely low space requirements, is lost.  See Q1 below.

---

> ### Author Response · Authors · 2022-08-02
> **Response to Reviewer pYST**
>
> We appreciate the valuable reviews. There are four concerns from the reviewer, so we would like to address each below.
>
> #### **More motivation to the time-varying covariance matrix setting**
> The time-varying covariance environment occurs in any linear dynamical system (we assume linear here to keep the Gaussian assumption) which evolves with time. Examples include the power grid (see the introduction section on this - Line 17-23), financial stock market, etc. Our motivation is precisely to study the evolving dynamics of such systems because these are typically governed by very few underlying modes.
>
> #### **More motivation to the objective (eigenspace for the last element in the data stream), compared with the top-k eigenvectors of the expectation of the sample covariance**
> We do agree with the reviewer that the expectation is a good performance measure with a clear interpretation for the iid case, however, its extension to the case of time-variant covariance remains unclear, especially the choice of distribution with which the expectation is taken. The considered objective is essential for many applications where the future prediction is more important than the regret at the present due to the uncertainty of the distribution. We have also included an extension of our results to a similar cumulative performance metric in Appendix A.
>
> #### **Analysis of robust streaming PCA with effective rank**
> We thank the reviewer for bringing this idea to our knowledge. For now, all our results rely on the spectral gap, and extensions to effective rank seem to require a new set of tools. We have updated the manuscript to reflect this as future work and included the necessary references (Lines 360-362).
>
> #### **Gaussianity assumption dependency for the minimax result**
> The lower bound result crucially relies on the Gaussianity assumption because we bound the KL-distance between two distributions. Consequently, these rates are conservative because of the dimension-dependence induced by the Gaussianity assumption. Further, as the reviewer correctly points out, we don’t need the Gaussian assumption when deriving the upper bound with Assumption 1, but the minimax is achieved when the Gaussianity assumption holds.

---

### Official Review · Reviewer_UR4D · 2022-07-11

**Rating:** 6
**Confidence:** 2
**Soundness:** 2 fair
**Presentation:** 3 good
**Contribution:** 2 fair

**Summary:**

This paper presents a novel streaming principal component analysis (PCA) method. A major novelty in the model is that the it investigates a robust perspective, where the covariance matrix belongs to a temporal uncertainty set.

The contributions are two-fold. On one hand, the paper provides a fundamental lower bound for estimating the principal components within the considered framework, namely when the covariance matrix belongs to a temporal uncertainty set. On the other hand, the authors provide an analysis for two algorithms to recover the principal components extensively used in the standard streaming PCA setting.

**Questions:**

See Weaknesses, mainly related to positioning with respect to the state of the art and weak experimentations.

**Limitations:**

See Weaknesses

**Strengths And Weaknesses:**


The paper is relatively well written, and the contributions are clearly identified.

A major issue of this paper is its position within the state of the art. While the authors do cite some relevant related methods, it seems that the authors either consider robust PCA literature, or streaming PCA literature, but never both frameworks.

Another issue with respect to the state of the art is that the authors provide some critics wrt related works [46, 11, 50, 25], as they consider “Oja’s algorithm … and do not provide a benchmark comparison”. However, this is exactly what the present paper does: it presents an Oja’s algorithm, and does not provide a benchmark comparison.

Experiments are not comprehensive. It is missing any analysis with a comparative study, neither in the main paper nor in the appendix. It is also missing some benchmark on real datasets that allows to clearly motivate the proposed work.

There are many spelling and grammatical errors, such as “it satisfies mild condition”, “the columns form basis”, “the ith largest eigenvalue “, “and and”, “we assume … and then draws”, “it is hard to compute of the“, “Note that if the we…”, “These feature models the behavior”, “these hypothesis”, “We consider following algorithms…”, “We used following parameters”,… and many others. A proofreading of the paper is required.

---

> ### Author Response · Authors · 2022-08-02
> **Response to Reviewer UR4D**
>
> We appreciate the reviewer giving clear directions to improve this paper. Here, we tried to resolve two weaknesses each. You can also find additional information on the standard review at the top and the revised submission.
>
> #### **Position of the present paper and other works in both robust and streaming PCA fields**
>
> Our contribution lies in proposing a new robust framework for the streaming PCA problem. As far as we know, there are no models or results that provide any guarantees (lower or upper) in this setting. We have updated our related works section to highlight this fact with more comprehensive references, including settings like masked features and sparse outliers. Please check the related works section of the new submission.
>
> #### **Real dataset experiments**
>
> In the revised manuscript, we provide a new experiment on the real-world S&P500 stock price dataset. The results support our main theoretical result: the existence of optimal parameters when the covariance is time-variant, and that block size (for noisy power method), and inverse learning rate (for Oja’s algorithm) play a similar role in the non-stationary environment. Further, due to the absence of any antecedent literature that considers streaming PCA under the proposed robust model, we do not have a benchmark to compare against. However, we would like to emphasize that the experiments on the real-world dataset validate our theoretical findings, which is the primary contribution of this work.

---

> > ### Comment · Reviewer_UR4D · 2022-08-08
> > **Rebuttal**
> >
> > I thank the authors for the reply. I also thank them for modifying the paper and for providing new experiments in the appendix of the paper.

---

### Official Review · Reviewer_i2FX · 2022-07-12

**Rating:** 7
**Confidence:** 4
**Soundness:** 3 good
**Presentation:** 3 good
**Contribution:** 3 good

**Summary:**

This work analysis streaming PCA algorithms in the time varying covariance model and shows that the noisy power method that relies on a batch of samples at each update is Reta-optimal, as opposed to Oja's method that relies on a single sample per update. The proposed theory deviates from the mainstream setting where the fixed covariance model is employed. The proposed theory sheds light onto the performance of streaming PCA algorithms in a setting where the covariance matrix changes over time. The proposed theory is demonstrated practically on synthetic data.

**Questions:**

1. Multiple redundant words and typos (e.g., 2 in line 38 instead of (2), "of" in line 133, "the" in line 136, "zt" instead of "wt" in line 224, period in line 318). Please go through the paper again and fix all inconsistencies.

2. Please rephrase lines 97-98. They are hard to follow.

3. Lines 135-139 present how quickly the At can vary. This is not mentioned earlier in the abstract or the introduction. Please mention this briefly in the abstract and/or introduction so that the reader is aware of this setting and is not misled into assuming no conditions on the rate of change of At.

4. Is the assumption 1 in definition 1 relevant in practice? Please provide examples.

5. Does Algorithm 1 need input batch size, B, with xt? If so, please add it to the pseudocode.

6. Does algorithm 2 require input learning rate, \zeta? If so, please add it to the pseudocode.

7. Line 301 refers to line 6 in Algorithm 2 but there are only 5 lines. Please edit this typo.

8. In the experimental studies, can we please see what happens for larger perturbation \Tau? Does the trend in Figure 1(a) hold for larger \Tau?

9. Please comment on the behavior of Oja's algorithm in Figure 1(a). It appears that larger learning rates are better for larger perturbations. Why? Please elaborate.


**Limitations:**

A limitation is method in the text and posed a future work.

**Strengths And Weaknesses:**

Strengths:
1. The paper is well-written and has a good flow.
2. The proposed theory is motivated with practical examples.
3. This work provided convergence analysis two important algorithms, namely the noisy power method and Oja's method under time varying covariance model.
4. The proposed theory in corroborated by synthetic studies.

Weaknesses:
1. Multiple redundant words and typos.
2. Some of the text is hard to follow.
3. Some analysis of the synthetic studies is missing.

---

> ### Author Response · Authors · 2022-08-02
> **Response to Reviewer i2FX**
>
> We thank the reviewer for their detailed comments and feedback. In addition to common questions addressed before, we provide a pointwise argument below:
>
> #### **Practicality of Assumption 1**
>
> Assumption 1 implies that the sample covariance matrix's magnitude, expectation, and variance are bounded. This assumption always holds in data generated by most non-stationary dynamical systems such as the power grid because the quantity of interest such as the voltage magnitude, etc is finite. Further, researchers are often interested in studying slower variations of such systems, therefore the changes are smooth and have bounded moments. From a purely technical perspective, we would like to point out that our upper bound proof will not break under the slightly relaxed version of Assumption 1.
>
> #### **Experimental study on larger perturbations**
>
> We provide new experiments with larger perturbations in Appendix I.4 now. The experiments retain the trend of optimal values of the block size (learning rate) that are established in the main text.
>
> #### **Benefit of a larger learning rate on Oja’s algorithm for larger perturbations**
>
> Intuitively, the bigger learning rate can easily catch up with time-variant covariances while a smaller learning rate relies too much on the past samples and tracks the remote history.  The learning parameter balances the reduction in the stochastic error (where the larger block size is better) and error in the data stream from the distributional shift (where the smaller block size benefits). In the presence of non-stationarity, the algorithm needs to be sensitive to distributional shifts, so a larger learning rate is preferred.

---

> > ### Comment · Reviewer_i2FX · 2022-08-10
> > **Response to author reply**
> >
> > Thank you to the authors for responding in a detailed manner. I am convinced by the answers and recommend publication of the paper.

---

### Author Response · Authors · 2022-08-02
**Overall Response**

**We appreciate all reviewers for providing detailed and comprehensive feedback on our work.**

### **Typos & Soundness & Proof Readability**
We have fixed all typos and redundant words; rephrased and corrected those unclear parts as mentioned in this review. Since we submitted the new version, we addressed the new line number for each reviewer when we answered their concerns. We have also provided the notation table in the first part of the Appendix and rewrote some unclear parts throughout the proof in the appendix.

### **Experiment on the real-world environment**
There were some concerns about the practical effectiveness of our theoretical analysis since our findings were only verified in the synthetic experiments. Furthermore, some reviewers also recommended real-world experiments to strengthen our argument. Heeding to reviewer concerns, we have conducted simulation experiments using real-world data that corroborate our theoretical results. Our numerical results show that our theoretical results remain valid in the real-world datasets (details below and in Appendix J on the revised paper). However, since there are no existing works that consider our model, there are no algorithms that can numerically benchmark these results.

For the additional experiment, we are conducting the experiment on the S&P500 stock price dataset (see details in Appendix J). We first selected the companies, which have the cost data from Mar.18, 1980, to Jul.21, 2022. Then, we considered the 133-dimensional vector consisting of (normalized) daily return. Our task is to predict the principal components of the covariance matrix of daily returns as in [1-3]. Since the stock market is trivially non-stationary, our robust streaming PCA provides both modeling and algorithmic solutions to detecting underlying principal components.

### **Additional related works on the fields and citations**
We have cited existing work at the intersection of robust and online PCA (including Oja’s) algorithm. However, to the best of our knowledge existing work on robust subspace detection or Oja’s algorithm doesn’t consider time-varying covariances. Furthermore, as far as we know, there are no works that consider the proposed robust model or provide algorithms that deal with this non-stationarity.



[1] Yang, Libin. “An Application of Principal Component Analysis to Stock Portfolio Management.” (2015).

[2] Fatima, Samreen & Fraz, Tayyab & Shafi, Rfia. (2019). The application of principal component analysis and factor analysis to stock markets returns. 10.21474/IJAR01/9007.

[3] Souma, Wataru. (2021). Characteristics of Principal Components in Stock Price Correlation. Frontiers in Physics. 9. 602944. 10.3389/fphy.2021.602944.


**(Update at Aug 4, 3:00 AM UTC / Aug 8, 2:30 PM UTC)**
We now offer the source code for real-world experiments. Please refer to the recent supplement. Additionally, we updated Figure 5 (repetition was 3 in the previous version, now it is 5 as we indicated) and fixed small errors in the revised manuscript.

---

### Meta-Review · Area_Chair_DYZV · 2022-08-27

**Recommendation:** Accept
**Confidence:** Certain

**Metareview:**

The authors consider the streaming principal component analysis problem where the data generation process may be subject to perturbations. The paper provides fundamental limits and analyze the convergence of the noisy power method and Oja's method in this setting. This shows that the former is rate-optimal. The reviewers found the paper to be a solid contribution. Further, the paper has undergone some substantial improvements during the discussion process, which I commend the authors on. I recommend acceptance while strongly encouraging the authors to take any remaining reviewer comments into account while crafting the next version of the manuscript.

**Award:**

No

---

### Decision · Program_Chairs · 2022-09-14

Accept